



# Concentration and variability of ice nuclei in the subtropic, maritime boundary layer

André Welti[1], Konrad Müller[1], Zoë. L. Fleming[2], and Frank Stratmann[1]

[1]Leibniz-Institute for Tropospheric Research (TROPOS), Leipzig, Germany
[2]National Centre for Atmospheric Science (NCAS), Department of Chemistry, University of Leicester

*Correspondence to:* A. Welti (welti@tropos.de)

**Abstract.** Measurements of the concentration and variability of ice nucleating particles in the subtropical, maritime boundary layer are reported. Filter samples collected in Cape Verde over the period 2009-2013 are analyzed with a drop freezing experiment sensitive to detect the few, rare ice nuclei active at low supercooling. The data-set is augmented with continuous flow diffusion chamber (SPIN) measurements at temperatures below -24 °C from a two month field campaign at Cape Verde

in 2016. The data set is used to address the questions: What are typical concentrations of ice nucleating particles active at a certain temperature, what affects their concentration, what is their composition and where are their sources?

To investigate what the most common ice nuclei are and to identify the sources, bulk chemical aerosol composition obtained from the utilized filter samples is tested for correlations with ice nuclei concentration. It is shown that no significant correlation between the rare ice nuclei and the bulk aerosol chemical composition, which could serve as tracer for a specific aerosol class

e.g. of maritime origin, can be made.

Concentration of ice nuclei is found to increase exponentially with decreasing temperature. It indicates that several groups of particles with different ice nucleation properties (size, composition) are contributing to the ice nuclei concentration at different temperatures. The concentration of ice nuclei active at a specific temperature varies over a wide range. The frequency with which a certain ice nuclei concentration is measured within this range is found to follow a log-normal distribution. The log-

normal frequency distribution can be explained by random dilution associated with turbulent, long-distance transport.

To investigate the geographic origin of ice nuclei, source attribution of air masses from dispersion modeling is used to classify the data into 7 typical situations. While no source could be attributed to the ice nuclei active at temperatures higher than -12 °C, concentrations at lower temperatures tend to be elevated in air masses originating from the Saharan desert.

## 1 Introduction

Ice crystals form in many ways in the atmosphere. If ice formation initiates on ice nucleating particles (IN) immersed in supercooled cloud droplets, the ice nucleation mechanism is named immersion freezing. The effect immersion freezing exerts on cloud extend, lifetime, radiative properties and rain formation depends on the concentration of IN active at a particular temperature (DeMott et al., 2010). Very high IN-concentrations above $100\,L^{-1}$ lead to the formation of numerous small floating crystals known as diamond dust, whereas low concentrations e.g. $0.01\,m^{-3}$ would produce large ice crystals falling from

a cloud in distances meters apart. Satellite observations (e.g. Carro-Calvo et al., 2016) and ground-based remote sensing



(Ansmann et al., 2009) show that above -10 °C ice containing clouds are rare and often cloud top temperatures below -20 °C are necessary for clouds to glaciate.

Glaciation of supercooled clouds often initiates at the cloud top where ice is formed by immersion freezing (primary ice formation). Ice crystals growing at the expense of cloud droplets (see e.g. Korolev, 2007) settle by gravitation and upon contact

with supercooled droplets, secondary ice formation mechanisms (most efficient at -5 °C) multiply the ice crystal concentration within the cloud above the number of IN present (e.g. Heymsfield and Willis, 2014; Hobbs and Rangno, 1985; Mossop et al., 1970). Conditions within marine clouds make them susceptible to ice multiplication processes (Heymsfield and Willis, 2014) so that already few active ice nuclei could lead to the formation of many secondary ice crystals. At what concentration ice crystals exert significant influence on the properties of the cloud in which they form has been addressed by Rangno and Hobbs (1988).

From aircraft and mountaintop observations Rangno and Hobbs (1988) identified the significant ice crystal concentrations able to produce precipitation to be on the order of $1\,L^{-1}$ or more, cf. Fig. 1. In their data-set, such concentrations have been measured in clouds with top temperatures between -5 °C and -10 °C.

Satellite data (Carro-Calvo et al., 2016; Rosenfeld et al., 2011) and aircraft observations (Rangno and Hobbs, 1991, 1994) agree on a land-sea contrast with the tendency of cloud glaciation at higher cloud top temperatures over sea. Carro-Calvo et al.

(2016) offer the explanation that the presence of larger sized droplets in maritime clouds, which are required for effective secondary ice formation (Heymsfield and Willis, 2014) could play a role.

Several studies (e.g. Hobbs and Locatelli, 1970; Bertrand et al., 1973; Borys and Duce, 1979; Castro et al., 1998) observed that IN-concentrations in maritime air masses are generally lower than in continental air, suggesting no important source for IN from the ocean. But Bigg (1961) or Soulage (1961) measured an increased IN-concentration in air of maritime and coastal

origin, providing directly conflicting evidence for the importance of the ocean as IN source.

Two main sources for IN in maritime air have been proposed. Long-range transported continental aerosol (mainly dust) suggested by Bigg (1973) or marine organic ice nuclei of biogenic origin aerosolized with the sea spray (Schnell and Vali, 1975; Rosinski et al., 1987). Aerosol from both sources have been investigated (e.g. DeMott et al., 2003, 2016, and references therein) and found to provide efficient IN. From laboratory experiments it is established that dust particles tend to nucleate

ice below -20 °C whereas biological particles can initiate immersion freezing at temperatures up to -5 °C. Joly et al. (2014) demonstrated that the particles initiating immersion freezing in cloud water samples collected at Puy de Dôme consist to an increasing fraction of biological IN (identified by sensibility to heat treatment) towards higher freezing temperatures. They estimate 77% biological IN at -12 °C increasing to 100% at -8 °C.

Beside the different temperature range in which dust or biological particles act as IN, they can generate a specific signature

in the concentration-temperature spectra (cf. Sec. 4 and Appendix A) by which they can be identified. Biological particles and in particular bacterial IN of one source tend to exhibit homogeneous ice nucleating properties. They initiate ice formation in a narrow temperature range seen as step like increase in concentration in a temperature spectra (e.g. Murray et al., 2012). In contrast, dust particles activate in a broader temperature range (e.g. due to the influence of particle size on nucleation efficiency) seen as exponential increase in concentration towards lower temperatures (Bigg, 1961). If a strong marine source of biological





origin exists it can be expected to be detectable as an inflection in IN-concentration at temperatures above -16 °C where the ice nucleating fraction of dust particles is small.

The concentration of potential IN measured at a certain location and temperature undergoes rapid changes (Bigg, 1961; Bigg and Hopwood, 1963). This can be seen as indication that in terms of IN the atmosphere is poorly mixed. Pockets of high 5 concentration can be followed by an almost IN free period as air moves by the location of measurement.

The two main factors influencing IN-concentration and variability at a certain location are:

1. the characteristics of the present population of aerosol particles (e.g. size-distribution, composition)

2. transport processes (initial concentration of the IN at its source, location of the source relative to the sampling location, modification and dilution during transport (Anderson et al., 2003)).

10 Membrane filter samples have been used to detect ice nuclei at low supercooling since the beginning of ambient IN-concentration measurements (Bigg, 1961; Bigg et al., 1963). Long sampling times are necessary to collect enough air volume to be sensitive to the low concentration of rare IN able to initiate immersion freezing at low supercooling (above -20 °C). Some problems of long sample exposures were identified by (Mossop and Thorndike, 1966). The collected particles are subject to gas condensation, aging or can be covered by aggregating smaller particles during sampling which could reduce the concentration 15 of active IN. Daily integrated samples not only average out minor erratic fluctuations but also potential high count periods (e.g. due to the dissipation of the nocturnal inversion) within the sample time. Higher IN-concentrations (up to three orders of magnitude, cf. Sec. 8) than the reported background concentration can be expected on timescales smaller than what is resolved by the used filter collection.

## 2 Sampling site

20 Samples are taken at the Cape Verde Atmospheric Observatory (CVAO; 16,848 °N, 24.871 °W) strategically placed at the northeastern shore of São Vincente (one of the northern islands of the Cape Verde archipelago). Cape Verde islands are located 16 °N, 570 km off the coast of West Africa. Situated in the trade wind zone the northeasterly winds prevail throughout the year, bringing in air masses from the open ocean. São Vincente is downwind of the coastal upwelling region on the west coast of Africa with high marine biological productivity and the outflow of the Saharan desert. Characteristic aerosol particles are of 25 marine origin (e.g. sea-spray) mixed with a continental background (mineral dust and smoke from biomass-burning), (Fomba et al., 2014). From Dec-Feb an usually strong Saharan desert dust period is typical. Filter samples are collected on a tower 30 m above the sea surface, 70 m from the coastline. The 30 m tower reaches out of the high sea salt loaded ground layer (Blanchard and Woodcock, 1980) into the mixed, maritime boundary layer.

## 3 Experimental method

30 150 mm quartz fiber filter, sampled using a Digitel filter sampler (DHA-80) with a $PM_{10}$ inlet are used to obtain a time series of ice nuclei concentration with 1-3 day resolution. The $PM_{10}$ inlet excludes particles $\geq 10 \mu m$ from being sampled. To determine



the IN-concentration, 103 random sub-samples, each containing aerosol from an air volume of 37-114 L are punched out of each filter, immersed in 100 μl droplets inside 0.5 ml safe-lock tubes and subject to temperatures down to -25 °C, allowing ice formation by immersion freezing. Freezing by impurities in the water typically sets in below -20 °C. Blank sub-samples taken from filter through which no air was drawn, start to cause freezing below -16 °C. It is not before -20 °C that the filter material

initiates ice formation in 50% of the samples. The drop freezing assay is set up following the description in Conen et al. (2012), with the difference that the circular sub-samples used for the present measurements are only 1 mm in diameter. According to Vali (1971), the cumulative concentration of IN per air volume as a function of temperature ($K(\theta)$) can be calculated by

$$K(\theta) = (\ln N_0 - \ln N(\theta))/V, \tag{1}$$

with $N_0$ denoting the number of sub-samples, $N(\theta)$ the number of unfrozen sub-samples at temperature $\theta$ and $V$ the volume

of air passed through each 1 mm sub-sample. With the given number of sub-samples, their diameter and sampled air volume the method is sensitive to concentrations from approx. 0.08-130 IN/m$^3$. Following Conen et al. (2012) the uncertainty (one standard deviation) in measuring $K(\theta)$ is given by

$$dK(\theta) = \pm\sqrt{K(\theta)}/(N_0 \cdot \sqrt{V}). \tag{2}$$

For clarity of presentation, data in the figures below are shown without the range of uncertainty given in Eq. (2).

During a field campaign in Jan.-Feb. 2016 concentration measurements of IN active at low temperatures, were conducted using the SPectrometer for Ice Nuclei (SPIN). SPIN is a parallel plate continuous flow diffusion chamber designed based on the chamber discussed in Stetzer et al. (2008). Aerosol was sampled at a flow of 1 L/min through an inlet up on the same 30 m high tower where the PM$_{10}$ filter sampler was located. An impactor (TSI, 0.071 cm orifice) was used to limit the particle size sampled with SPIN. For details on the SPIN instrument we refer to the description in Garimella et al. (2016)

## 4 Temperature spectra of cumulative nuclei concentration

The temperature spectra (Fig. 1) summarizes data from 500 analyzed filter samples which were collected from 2009 - 2013 at the CVAO. Data in red, green and blue represent measurements at -8 °C, -12 °C and -16 °C. These data are shown in the same colors in following Figs. 2, 3, 4, 5, B1, C1. 30 min average concentrations measured with SPIN during the intense campaign in 2016 complement the data set in the temperature range from -24 °C to the onset of homogeneous freezing below -38 °C.

The range of measured IN-concentration at constant temperature is large. Measurements vary up to three orders of magnitude for the 24 to 72-h filter samples and cover up to four orders of magnitude in the 30 min SPIN data. For the filter data, the temperature dependent change in concentration is similar to the slope of the approximation proposed in Fletcher (1962), indicated in Fig. 1. Concentrations measured with SPIN at lower temperatures show a weaker temperature dependence, vary in a larger range and diverge from the Fletcher approximation (cf. Fig. 1).



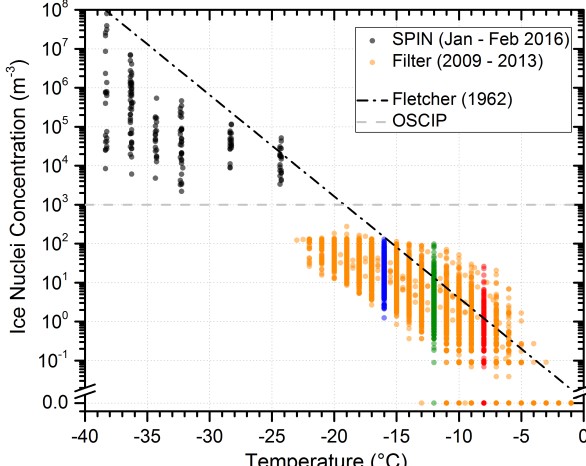

**Figure 1.** Cumulative ice nuclei concentration as function of temperature. Date and method of measurement are indicated in the figure. Filter data at -8 °C, -12 °C and -16 °C are color coded red, green and blue. This data are shown in the same colors in subsequent figures. The Fletcher (1962) approximation is given for comparison and the OSCIP (Onset of Significant Concentrations of Ice Particles, Rangno and Hobbs, 1988) indicates a guideline concentration for effecting cloud properties by primary ice nucleation.

It seems indistinguishable if few particles with a high probability to act as IN or many less active aerosol particles represent the concentration of IN at a certain temperature. From the cumulative temperature spectra, it can be argued that the former is the case. A low activated fraction of an abundant IN at one temperature would generate a steep increase in concentration with decreasing temperature in a narrow range (cv. Appendix A for an extended discussion on the interpretation of IN temperature spectra). Concentrations measured at CVAO are observed to increase exponentially with supercooling, indicating a broad variety of particle properties (e.g. size, composition, ice active surface sites) responsible for ice formation at different temperatures. The temperature dependent increase in concentration (slope of the temperature spectrum) is not constant for different filters, indicating active IN of varying nature and abundance that contribute to the temperature spectra at different occasions. For additional discussion of the variability in slopes we refer to Bigg (1961) and Appendix A. The scatter in IN-concentration at a temperature of interest is caused by variation in concentration of IN with a certain property in the sampled air mass. If properties of IN, active at a certain temperature, and therefore measured IN-concentration can be correlated to bulk chemical aerosol composition or air mass origin is analyzed in Sec. 6 and 7.

## 5   Time series

The variation of ice nuclei concentration with time is exemplary shown for -8 °C, -12 °C and -16 °C in Fig. 2. The four year during which filter samples were collected represent a cross section of the different atmospheric conditions (dust events, rain, dry-tropical) encountered on Cape Verde. Neglecting extreme values, IN-concentration is changing within two orders of mag-



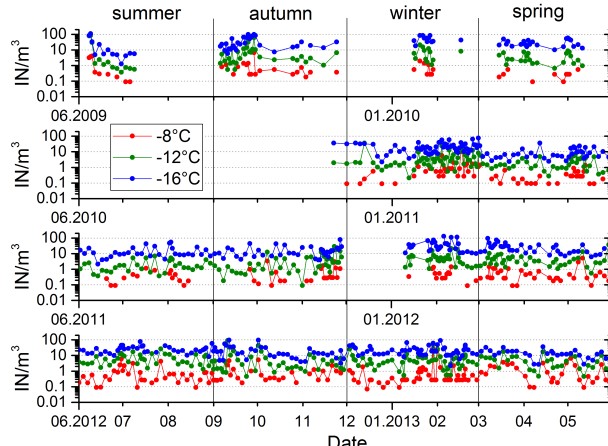

**Figure 2.** Four-year time series of ice nuclei concentration at three temperatures indicated in the figure.

nitude, without obvious cycles in the series. The measurement in the time series are positively first-order autocorrelated for all temperatures between -6 °C to -16 °C, i.e. persistence in above/below average IN-concentration on consecutive filter samples. Concentrations measured on samples further apart are not autocorrelated. Variations in IN-concentration are non-synchronous at different temperatures (cf.Fig. 2), indicating differences in origin of the ice nuclei populations active at different temper-

5 atures. This observation is analog to the change in slope of the temperature spectra for different filter samples discussed in Sec. 4. With increasing difference in temperature, the variation in IN-concentration at two temperatures become less correlated ($R^2$=0.30, $R^2$=0.37, $R^2$=0.09 for -8 °C to -12 °C, -12 °C to -16 °C, -8 °C to -16 °C). The time series shows no distinct seasonal or inter-annual trends during the four years of measurements (see Appendix B for an inter-annual comparison of season separated IN-concentrations).

## 6 Bulk chemical composition

The 24 to 72-h filter samples were used to analyze the bulk chemical composition of the collected aerosol. Methods and details of the chemical analysis can be found in Fomba et al. (2014) where the results of the chemical analysis from samples collected between 2007 - 2011 at CAVO are presented. Besides the derivation of the $PM_{10}$ aerosol mass the chemical analysis included ion mass concentrations of $Na^+$, $NH_4^+$, $K^+$, $Mg^{2+}$, $Ca^{2+}$, $Cl^-$, $Br^-$, $NO_3^-$, $SO_4^{2-}$, $C_2O_4^{2-}$, organic (OC) and elemental carbon

(EC).

The sampling time allows to resolve aerosol mass concentration and composition on a synoptic scale (on the order of $1000\,km$). It is assumed in this sampling strategy, that aerosol composition mainly changes with horizontal changes in air masses (e.g. frontal systems) that occur on this scale. Additionally, variation in bulk aerosol mass concentration and composition on the synoptic scale can often be related to sources via back trajectories (Anderson et al., 2003). If IN sources were related to the





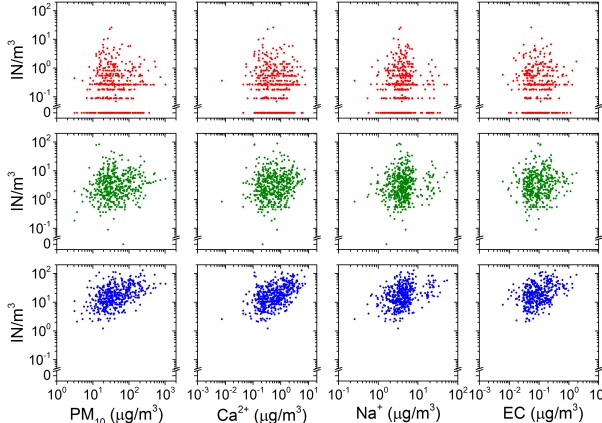

**Figure 3.** Relation between the concentration of IN at -8 °C (red), -12 °C (green), -16 °C (blue) and total particle mass concentration, mass concentrations of calcium ($Ca^{2+}$), mass concentration of sodium ($Na^+$) and mass concentration of elemental carbon (EC). $Ca^{2+}$, $Na^+$, EC are tracers of continental, marine and combustion sources, respectively.

sources of bulk chemical components, they could be used as tracer to identify source region or aerosol constituents contributing to the IN population. Fig. 3 shows scatter-plots of IN-concentration at -8 °C, -12 °C and -16 °C versus $PM_{10}$ as tracer for desert dust and concentrations of $Ca^{2+}$, $Na^+$, EC as indicators of continental, maritime and combustion sources. Except an insignificant tendency ($R^2$=0.11, $R^2$=0.17) of higher IN-concentrations at -16 °C towards higher $PM_{10}$ and $Ca^{2+}$ mass
concentration, the scattering is uncorrelated. Using Spearman statistics, monotonic association (more is more or more is less) between the IN-concentration at temperatures from -5 °C to -16 °C and mass concentration of all chemical components is tested. No significant correlation exists. While the IN-concentration at temperatures above -16 °C is not correlated to the amount of continental aerosol, evidence for the importance of the Sahara desert as source of IN at lower temperatures is found in the analysis of air mass origin (Sec. 7). This apparent contradiction can be explained as a non-constant ice active fraction of
Saharan dust aerosol. A variable ratio of dust to IN-concentration has previously been observed by Bertrand et al. (1973) who measured in West Africa. A comparison of IN-concentration to the abundance of aerosol particles larger than a certain size can be found in Appendix C.

## 7    Effect of air mass origin

The origin of air mass sampled from 2009-2013 on the filter is determined using the NAME dispersion model (Jones et al.,
2007) and classified into 7 categories (as shown in Carpenter et al., 2010). NAME dispersion footprints are calculated for the air arriving at the site, showing where the air masses had been in the surface layer (0-100m) in the previous 10 days. For each footprint, the proportional residence time in each geographical region (Atlantic, North America, Europe, coastal Africa and North Africa) is used to classify each period into one of the 7 air mass categories as shown in Fig. 4. Concentrations measured



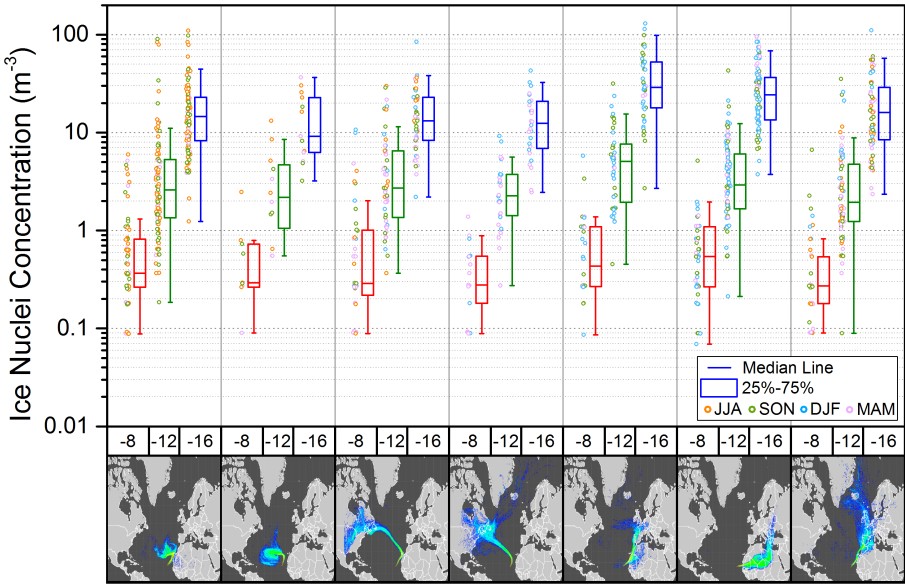

**Figure 4.** Variation of ice nuclei concentration classified into 7 situations of air mass origin at three temperatures. Data shown right of each box plot is color-coded by season (JJA=Jun-Aug, SON=Sep-Nov, DJF=Dec-Feb, MAM=Mar-May). Air mass classification from left to right: 1. African coastal, 2. Atlantic marine, 3. North American, marine and coastal, 4. North American and marine, 5. dust and Europe, 6. dust (e.g. Saharan and Sahel region), 7. coastal and Europe.

under the different categories at -8 °C,-12 °C and -16 °C are shown as box plots in Fig. 4. Slightly elevated concentrations are found for dust (from the Saharan and Sahel region) and Europe influenced air mass. The influence of dust on IN-concentration increases towards lower temperatures, where more dust minerals become ice active. Although air mass origin is correlated to season (dust storm season from Dec-Feb), no seasonal trend in IN-concentration is found. Which ice active aerosol Europe contributes is unclear. It could be of industrial or agricultural origin. From the comparison of the 7 situations categorized in Fig. 4 it is seen that continental effected air masses contain higher IN-concentrations than pure maritime air mass. Vice versa, even though the location of measurement is in vicinity to high primary oceanic productivity no correlation of high IN-concentration with coastal and marine air mass have been found in the temperature range accessible by the drop freezing method.

## 8 Frequency distribution

The IN-concentration and variability measured in one location depends on the nature of the IN sources (strength and ice nucleation efficiency) and modification during transport (aging and dilution). While the cumulative temperature spectra (discussed in Sec. 4) can provide information on the efficiency and abundance of IN, the frequency distribution can be used to investigate





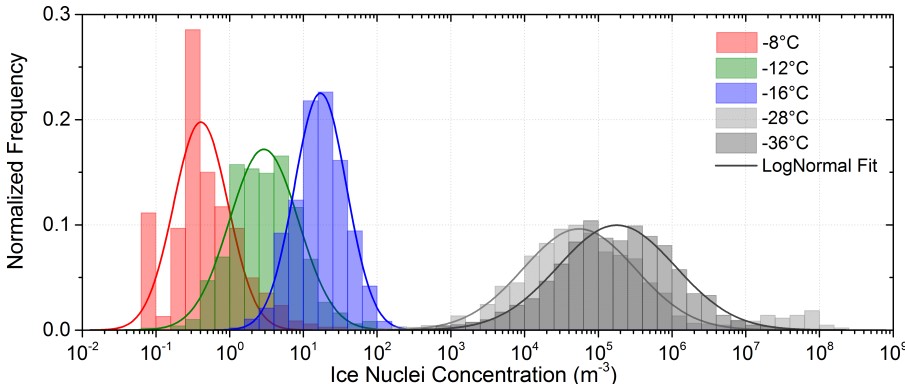

**Figure 5.** Frequency distribution of measured IN-concentration with log-normal fit curves.

the effect of transport.

It is found that log-normal frequency distributions best approximate the measured variability in concentrations at each individual temperature (cf. Fig. 5). The suitability of a log-normal distribution to approximate IN-concentrations frequency distributions has been recognized previously by Maruyama (1961); Isaac and Douglas (1971); Radke et al. (1976). Isaac and Douglas (1971) demonstrated that the frequency distribution of IN-concentration does not follow a Poisson distribution, thereby proving that IN are not randomly distributed in the atmosphere.

An explanation for the log-normal frequency distribution in concentration (such is also often found in concentrations of ambient pollutions) has been given by the theory of successive random dilutions (Ott, 1990). A log-normal distribution indicates that the initial IN-concentration at the source has undergone a series of random dilutions while being transported. In the case of IN-concentration where ice nucleating particles are lofted from the surface and transported in the free atmosphere to the measurement site, turbulent mixing randomly dilutes the initial concentration. Variation in source strength (e.g. particle concentration) does not change the log-normal standard deviation of the distribution, but causes a shift in concentrations. Only proximity of the measurement location to the IN source, and thus the maximum concentration possible (no dilution) would cause an even more skewed frequency distribution with a stronger downward bend at high concentrations (Ott, 1990). Neither the frequency distribution obtained from filter measurements nor the measurements with SPIN show this feature. The uni-modal, regular bell shape of the frequency distribution on the log-scale indicates the absence of a strong local source. Measured concentrations can therefore be assumed to represent the background concentration in a subtropic, maritime environment defined by long-range transport.

## 9 Discussion

The comparison of IN-concentrations from filter samples to chemical bulk aerosol composition did not allow to identify the nature or source of IN active at low supercooling. Potentially important particle classes are mineral dusts and biological parti-



cles advected with dust or from marine source. Based on laboratory measurements of the ice nucleation efficiency of Saharan dust from the Hoggar region, Pinti et al. (2012) suggested that during dust events sufficient dust IN active between -13 °C and - 23 °C could be present at Cape Verde to initiate cloud glaciation. Classification of source regions into 7 prevalent situations support the importance of continental IN by slightly elevated median IN-concentration during continental air mass situations.

From Lidar observations discussed in Ansmann et al. (2009) it is known that generally temperatures need to drop below -20 °C for glaciation to start in altocumulus clouds (altitude 7 km) over the Cape Verde. Cases when clouds glaciated at higher temperatures between -5 °C to -20 °C could often be attributed to seeding of the supercooled cloud with ice crystals from clouds at higher altitude. The present measurements of potential IN-concentrations at ground level indicate that glaciation can start at higher temperatures, but OSCIP (indicated in Fig. 1) would typically be reached at -20 °C.

The main problem to identify the composition or source of IN by correlating concentration measurements with information on bulk chemical composition must be the small fraction of IN in all particles. Only when an IN source contributes a unusual large fraction to the aerosol e.g. during a dust storm (Boose et al., 2016) or downwind of a forest fire (McCluskey et al., 2014), bulk chemical composition can corroborate the IN source. Efforts to pinpoint the contribution to IN-concentration from specific sources should focus on measuring close to the source or investigating collected samples under laboratory conditions.

No evidence was found that the ocean is a general source for high IN-concentration in the subtropics. This confirms observations by e.g. Isono et al. (1959); Carte and Mossop (1960); Hobbs and Locatelli (1970) that IN-concentration in air masses with a purely maritime trajectory tend to be low. Ice nucleating particles that can originate from sea water and are active at low supercooling (Schnell and Vali, 1975; DeMott et al., 2016) might be airborne in too low concentrations and in general only contribute as a minor source to the background concentration. However, low IN-concentrations must not indicate low ice
crystal numbers in developed clouds if conditions favor secondary ice multiplication mechanisms. Also in particular circumstances (e.g. algae bloom) the purely marine IN could be of regional importance. Burrows et al. (2013) suggest strong regional differences in the importance of marine biogenic and dust IN, with the highest impact of marine IN on cloud properties in the remote Southern Ocean far from strong dust sources.

Filter samples are collected on timescales of 1 to 3 day sampling time, resolving variations on a synoptic scale (order of
1000 km, typical for frontal systems). The sampling strategy is chosen based on the assumption that concentration and composition of aerosol are constant within the horizontal scale of the air mass passing the sampling spot in this time (Anderson et al., 2003). Using filter samples aiming for aerosol composition monitoring therefore provide insight into variation and concentration of IN on a synoptic scale. The sampled air volume and the drop freezing method confine the sensitivity of the measurement to the rare IN active at low supercooling. Consequently the present time series obtained from filter samples gives information
on the variation and concentration of highly active IN on a synoptic scale. From SPIN measurements (cf. Fig. 5) using a sampling time of 10 s (corresponding to a scale on the order of 100 m), high variability within the synoptic scale is observed. Although SPIN measurements are at lower temperatures where different particle types contribute to the IN population, the variability of rare IN counts, if it could be observed at high temporal resolution, could be of the same order as the variation in low temperature IN-concentration. Only due to the extend of the filter time series, the range of variability at the sample location
partly covers a range of variability that approaches the range observed in the much shorter time domain. This highlights the



need to collect large enough (long time-serie or high frequency) data sets to characterize variability and correlation to aerosol properties.

## 10  Conclusions

Typical IN-concentrations in the subtropic, marine environment are obtained from 500 particle filter samples collected over 4-year and from a 2-month field campaign with the SPIN instrument. Besides establishing typical concentrations in a broad temperature range, fluctuations in the continuous filter data set are compared to aerosol chemical information and airmass origin to investigate possible sources and probable chemical composition of IN.

The subtropic, marine ice nuclei concentration is highly variable on a synoptic scale and even more on a higher resolution. The frequency of measured concentration is approximately log-normally distributed, pointing to a major influence of turbulent dilution during long-range transport from the source on IN-concentration. Dilution during transport can effect the IN-concentration more than season, air-mass source region or bulk chemical aerosol composition.

Continental air mass contain higher IN-concentration than maritime air mass. The more marine air-masses are mixed with continental air the lower the IN-concentration. Consequently, mainly long-range transport of randomly diluted continental sources of IN account for the concentration in the cumulative temperature spectra. However, low concentrations of probable biogenic IN, active at T > -10 °C that could originate from the ocean are found.

Diversity of the ice nuclei population manifests in an exponential increase in concentration towards lower temperatures of the cumulative temperature spectra.

By comparing the IN-concentration at low supercooling to the aerosol bulk chemical composition it is demonstrated that neither air masses rich in biogenic nor desert dust aerosol generate a strong increase in the measured ice nuclei concentration. Consequently the small subset of aerosol active as IN from these sources is either not constant or different sources account for the fraction of ice nucleating particles in the maritime environment. The contribution of each source to the cumulative temperature spectra depends on the source strength, fractional mixing (according to the successive, random dilution theory) with IN free air and the temperature range to which the source contributes ice nucleating particles. Multiple sources of temperature specific activity together with random dilution create the appearance of an ubiquitous, almost constant background concentration of IN.

Irrespective of season and air mass origin, the frequency distribution of ice nuclei concentrations at a certain temperature can be described by a log-normal distribution. Parametrizations in numerical models should reproduce this feature.

No fixed ration between the concentration of dust, marine or any other particle type and IN-concentration was found. This is in agreement with Bertrand et al. (1973) who observed a varying ration between the concentration of dust and IN at -20 °C in West Africa.

Assuming 1000 $IN/m^3$ to be a threshold concentration (Rangno and Hobbs, 1988) to have a significant effect on properties of supercooled clouds, temperatures below the range covered by the drop freezing experiments need to be reached to have an effect. If the temperature dependent IN-concentration exponentially increases with the slope of the Fletcher (1962) approxima-





tion, a primary effect of ice nucleation by immersion freezing on cloud glaciation and precipitation formation can be expected at $-20\,°C\pm5\,°C$, matching the -20 °C cloud glaciation temperature over Cape Verde reported by Ansmann et al. (2009). The SPIN data measured at temperatures below -24 °C support the temperature trend. Given that measurements of IN-concentrations at ground are representative for higher altitudes, the observation of cloud glaciation at $-20\,°C$ where IN-concentrations reach

5   OSCIP support the importance of immersion freezing on cloud properties in a subtropical, marine environment.

*Acknowledgements.* We acknowledge financial support from the European Union's Seventh Framework Programme (FP7/2007-2013) project BACCHUS under grant agreement n°603445. We would like to thank the UK Met Office for use of the NAME dispersion model and the STFC JASMIN supercomputer for hosting the model. Collection of filter at CVAO was made possible thanks to support by the German

10   BMBF within the SOPRAN I and II projects (FKZ: 03F0462J and 03F0611J). Élise Beaudin, Nadja Samtleben, Lisanne Hölting, Mareike Löffler and Pit Strehl were principally responsible for preparation and conducting the filter sample experiments. Thomas Müller provided particle size distribution measurements from CVAO.

Edited by:





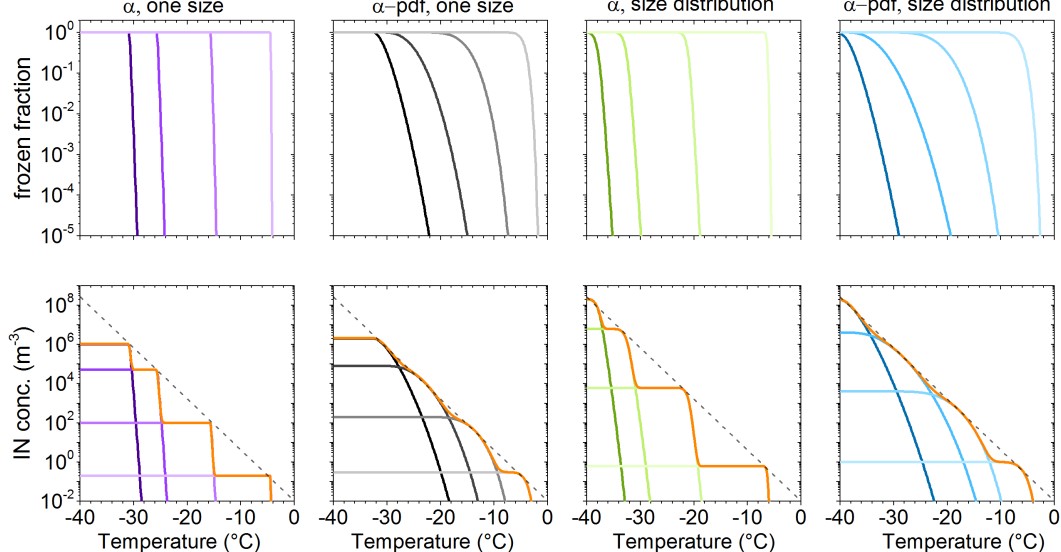

**Figure A1.** Illustration of the framework of the cumulative temperature spectrum. Each sub-figure in the upper row shows frozen fraction curves of four IN-groups with mean contact angles ($\alpha$) of $120°, 90°, 60°, 30°$. Above the sub-figures, variation in contact angle and particle size distribution within the IN-groups are indicated. A normal-distribution of contact angles (with $\mu = \alpha$, $\sigma = 6°$) and a log-normal particle size distribution are used to visualize the effect of variability within the four IN-groups. The second row shows scaling of frozen fraction with IN-concentrations per group chosen so curves touch the exponential parametrization of Fletcher (1962).

## Appendix A: Temperature spectra of cumulative IN-concentration

Four exemplary temperature spectra, produced using classical nucleation theory to describe immersion freezing are shown in Fig A1. A contact angle distribution ($\alpha$-pdf, e.g. Welti et al., 2012) emulates the variation of ice nucleation efficiencies within a macroscopic homogeneous (size, composition) population of particles. Including particle size-distribution describes the effect

5 of different surface area within a particle population of the same material. Comparing the four sub-figures in the top row of Fig. A1, it can be seen that the effect of a contact angle distribution on flattening the slope of the frozen fraction (FF) is stronger than particle size variation.

In the following we explore typical features found in temperature spectra. We refer to IN species with the same frozen fraction curve as IN-group. The number of IN in an IN-group scaled with its temperature dependent FF determines the IN-concentration

10 a group contributes to the temperature spectra. The number of IN as a function of temperature, i.e. the cumulative temperature spectra of an ambient measurement, consists of the IN-concentration from several IN-groups with partly overlapping activation temperatures (temperature where 0<FF>1).

Several IN-groups can sum up to an almost smooth exponential temperature spectra with no inflections (bumps). In this case the points of inflection in the frozen fraction of individual IN-groups (FF=0.5) construct the slope of the spectra (cf. Fig. A1 second



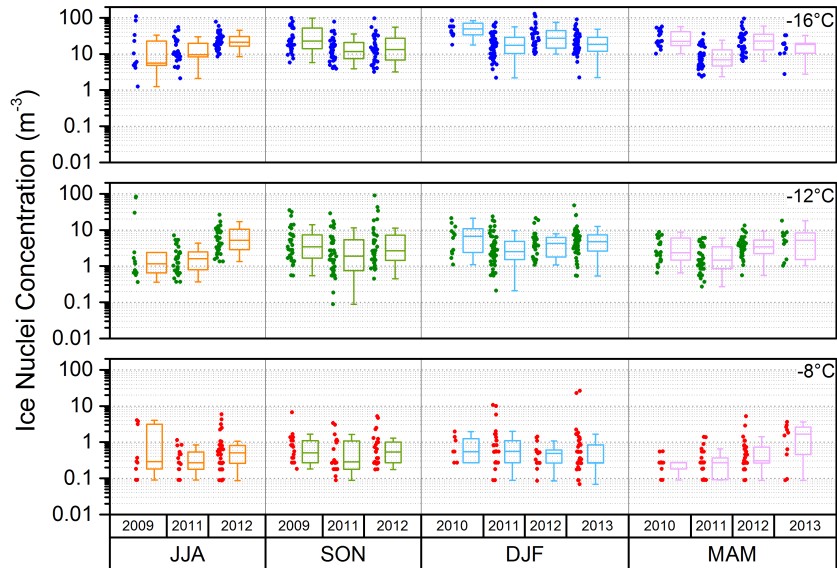

**Figure B1.** Seasonal variation in ice nuclei concentration separated for each year of measurement. (JJA=Jun-Aug, SON=Sep-Nov, DJF=Dec-Feb, MAM=Mar-May)

row). Reporting average freezing temperatures ($T_{50}$) of FF=0.5 from laboratory studies are therefore a well suited measure to transfer these results to ambient ice nucleation.

Fluctuations in the number of IN in different IN-groups (e.g. due to advection of incompletely mixed air parcels with different IN content, Bigg and Hopwood, 1963) lead to a variation in the slope of the temperature spectra or cause a bump (see below).

A multitude of temperature spectra make the spread in observed IN-concentration seen in Fig. 1 with the frequency distribution shown in Fig. 5.

Temperature spectra with bumps, from the combination of 4 IN-groups are shown as orange lines in the bottom row of Fig. A1. The more narrow the IN properties of an IN-group, the steeper the increase in FF and the more step like it appears in a temperature spectra. A broader distribution of IN properties decreases the slope. High concentration of a single IN-group

would emerge as step like peak over the other contributions. Note that a combination of 4 IN-groups with broad distribution of IN properties suffice to approximate an exponential temperature spectra.

Additional discussion of temperature spectra can be found in Bigg (1961); Bigg and Hopwood (1963) and Vali (1971).

## Appendix B:  Seasonal variation of IN-concentration

Fig. B1 shows the range of seasonal and inter-annual variation in IN-concentration during the measurement period. Although

air mass origin is correlated to season (dust storm season from Dec-Feb, cf. Fig. 4), no seasonal trends are found. Inter-annual variability within seasons are small during the four year of measurements.



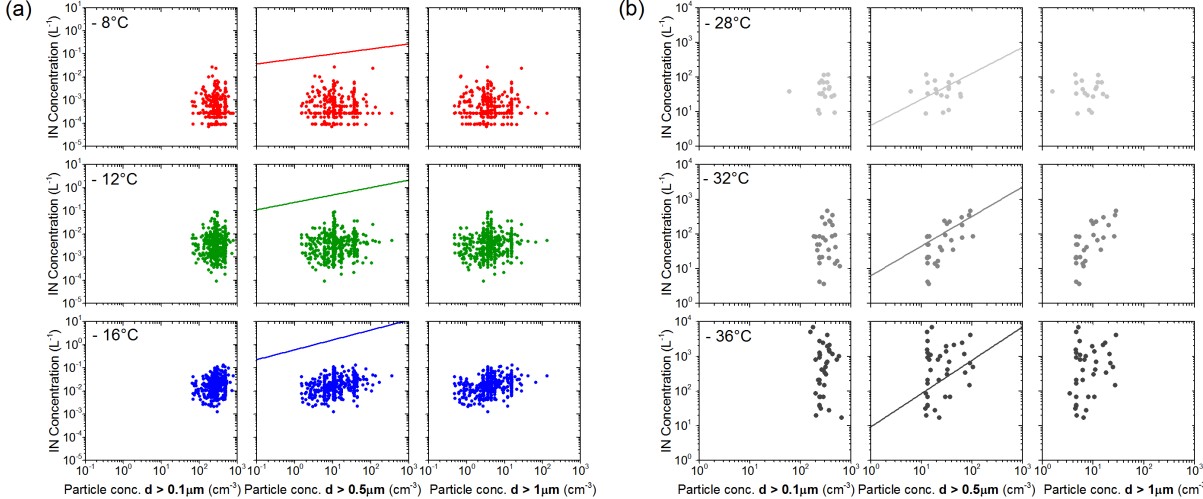

**Figure C1.** (a) 24 to 72-h average IN-concentration from filter experiments on the y-axis against total concentration of particle with diameter larger than 0.1μm, 0.5μm, 1μm on the x-axis, (b) 30min average IN-concentration from SPIN experiments versus total concentration of particles larger than 0.1μm, 0.5μm, 1μm. IN-concentration predicted by the DeMott et al. (2010) parametrization are given in the center row of (a) and (b).

## Appendix C: Comparison of IN-concentration to number concentration of aerosol particle with diameter larger than 0.1μm, 0.5μm, 1μm

IN-data in Fig. C1(a) at -8 °C, -12 °C, -16 °C are from filter measurements. IN-concentrations shown in Fig. C1(b) at -28 °C, -32 °C, -36 °C are measured using SPIN. The comparison is made for concentration of particles with diameter larger than 0.1 μm, 0.5 μm, 1 μm. IN- concentrations calculated according to the DeMott et al. (2010) parametrization which connects the concentration of aerosol particles above a threshold size to IN-concentration, roughly match the range of SPIN data in Fig. C1(b) within one order of magnitude. Discrepancy is largest for low particle concentrations. IN-concentrations from filter measurements are systematically lower than predicted by the parametrization. Note that the predicted concentration lies above the range of sensitivity of the filter method used in this study. Correlation of IN-concentration to the concentration of particles larger than a certain size is higher for 0.5μm and 1μm than 0.1μm. Non of the correlations is significant.



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
