# Peer review of "Concentration and variability of ice nuclei in the subtropic, maritime boundary layer"

_Atmospheric Chemistry and Physics, 2017_

## Referee Comment (RC1) · K. Bigg (Referee) · 4 Sep 2017

**Review of manuscript acp 2017-783, Concentration and variability of ice nuclei in the subtropic maritime boundary layer. A. Welti et al.**

**Keith Bigg, keith@hotkey.net.au**

This manuscript provides a potentially valuable long series of IN data for an area from which there have been no previous IN measurements that I am aware of. The topic is suitable for ACP and the methods used appear to be satisfactory, although there is a suspicion that concentrations may be underestimated. The language and presentation are clear and the few changes that seem desirable will be listed at the end of this review. This is a good paper and I hope that some of the following comments might be helpful. Where the authors disagree with those comments, I would be happy to enter into a debate

Summarising, I recommend publication after the authors have made any changes that appear to be appropriate.

1. **Abstract:** The paragraph that shows no relation between bulk chemical composition and IN concentrations is not very useful in the main text and certainly not in the Abstract. At a site where sea salt particles will be dominant by mass and IN active at temperatures relevant to cloud formation represent about one particle in a million, no other result would be credible.

2. **Introduction:** *p1., line 21*. Is there any firm evidence for concentrations as low as $0.01 m^{-3}$ except in polar regions? In addition to the listed factors influencing the formation of precipitation, updraft velocities and liquid water content could be added. Cloud extent suggests horizontal extent – cloud depth would be better.

3. *p2 line 1*: What concentration of ice crystals is necessary for ground-based detection? The presence of concentrations of IN in clouds at -10C will usually be similar to that of the concentrations in the air feeding the cloud of all biological IN capable of nucleation at that temperature. The list of potential IN from those sources continues to grow and concentrations must often exceed $100 m^{-3}$.

4. *P2. line 5*. Insert reference here: Hallett and Mossop, Nature 249, 26-28, 1974 who were first to investigate the effect experimentally and define the conditions necessary for it to operate. *Line 15* Hallett and Mossop defined that condition in 1974!

5. *P2. Line 19-22*. In Bigg's 1973 paper, 3 years of continuous measurements in the Southern Ocean, south Indian Ocean and south Pacific were summarised. The measurements were later used by Schnell and Vali (J. Atmos.Sci., 33(8), 1554-1564, 1976) to show that the measurements revealed a strong dependence on biological productivity. Their interpretation of biological IN rather than dust as the main factors in the measurements is much preferable to Bigg's. This work brings up an important point in relation to your manuscript. The ocean measurements were made with membrane filters that are known to undercount the concentration of IN in a salt-laden atmosphere, yet concentrations in the biologically productive zones were considerably higher than those reported in your manuscript. Chlorophyll measurements in the vicinity of the Cape Verde Is reported by Ramos et al. to indicate strong biological activity and significant biological IN should become airborne by bubble bursting. Am I right in assuming that your method only gives a "yes-no" answer

for the presence of an active IN at temperature T? If so it needs to be pointed out that actual concentrations may be higher.

5. *p.3 lines 3-4*. A spectacular example of the changes in IN concentrations that can occur was published in J. Meteorol. 15, 561-562, 1958. This was later interpreted to be due to related to enhanced biological populations resulting from heavy rain, with a proportion becoming airborne. (Atmos. Chem. Phys., 15, 2313-2326, 2015).

6. *p.3 line 11-12*. Long sampling times aren't necessary. Membrane filters with pore sizes 0.45μm or larger can be sampled at >10l/min but sampling >300l leads to serious undercounts. For long-term measurements or simultaneous measurements at many sites, sampling at 300l/day avoided logistic difficulties but averaged out any short-term fluctuations.

7. *p3. Line 14.* Reduction of the RH in the vicinity of a hygroscopic particle is a major factor. Allowing hygroscopic material from a $1m^3$ sample on a membrane filter to diffuse into an underlying wet filter, then drying the top filter and processing it, results in an IN count more than a factor of 2 higher than on a simultaneously sampled filter kept dry. (*This work has not been published – use the information if you want to*).

8. p.5, line2. At first I didn't understand this as all particles capable of forming IN at temperatures warmer than the test temperature will be activated. Does the answer lie in comment 5?

9. *Figure 2*. According to Ramos et al. chlorophyll is a maximum at the end of the year at Cape Verde. I don't see much evidence of a corresponding change in the -8C figure. As it is an important point in determining whether biological IN are effective at the site, running means of about 9 measurements at -8C shown on a diagram with a more extended scale might help. This procedure would be useful in reinforcing the surprising statement in lines 6 and 7 on p.8.

10. *Bulk chemical composition*. See comment 1.

11. *Air mass origin*. How many cases were involved in each of the 7 categories of figure 4? The sporadic rainy season from August-September would probably lead to much deeper atmospheric mixing at times and occasional scavenging of aerosol. How reliable are the 10-day trajectories to the site during that period?

12. *Frequency distribution, p.9 lines 7-9.* Size distribution of particles produced by a common method frequently have a log-normal size distribution and this can be expected from a local source. An alternative to Ott's random dilution hypothesis might simply be preservation of the original distribution during transport.

13. *p.10, line 6*. What is the minimum concentration of ice crystals needed for the lidar observations to detect them? It might be better to replace "to start" with "to be detected".

14. *Figure B1, p.14.* It might be interesting to have a separate diagram for the "rainy" season of August and September and for the period of maximum productivity, October-December.

**Minor typographical and construction errors.**

p.1 line 22 : change "cloud extend" to "cloud depth".

p.2 line 26. Change "consist to" to "consist of".

p.5 line 14 "exemplary shown". Change to "exemplified by".  Change "year" to "years".

p.8. line 6. Change "effected" to "affected".

p.9 line 20. Change "to identify" to "identification of".

p.10. line 10. Change "to identify" to "in identifying". Line 19: change "must not" to "need not".

---

## Referee Comment (RC2) · Anonymous Referee #2 · 6 Nov 2017

First of all, this is a very impressive set of measurements. The field of atmospheric ice nucleation is lacking long term data sets with which to compare models and test our understanding. Hence, a dataset comprised of 500 individual measurements over the course of 5 years is extremely valuable. Hence, I support its publication.

However, I think there are some aspects of the paper which need to be improved prior to publication. I go through these in detail below:

Specific comments:

1) P1. Ln 11. Why does an exponential change in INP concentration suggest that several groups of particles with different ice nucleating properties are contributing to INP populations? The exponential dependency on T could also just be explained by a

distribution of sites across a surface, rather than particles of different composition or size.

2) Consider using the term 'ice nucleating particles; INPs' rather than ice nuclei (IN). Vali et al. [2015] in their recent definitions paper came up with some compelling reasons why this is a better and less confusing term.

3) P1 ln 24. Provide a citation for 100 L-1 INP leading to diamond dust. My understanding was that diamond dust was a relatively low concentration of ice crystals of relatively large size, i.e. in contradiction to the statement made here. In addition, I understand diamond dust tends to form in clear air, without the presence of a liquid cloud.

4) P2, ln 1. When discussing data like that of Ansmann et al. and making statements such as 'above -10 C ice containing clouds are rare', make sure it is stated what sort of clouds are being referred to. For example, in convective clouds ice formation above -10oC is common. Ansmann et al. deal with shallow clouds.

5) P2. Ln 3. This paragraph is very confused. Parts of it seem to be referring to ice formation in shallow cloud types (e.g. stratus), whereas it then morphs into a discussion about secondary production which is more relevant for deep clouds.

6) P2. Ln 17-25. This discussion of marine INP is lacking reference to some more recent literature on the subject, e.g.: [Burrows et al., 2013; McCluskey et al., 2017; Vergara-Temprado et al., 2017; Wilson et al., 2015; Yun and Penner, 2013]. I appreciate the effort made to go back to much older studies, but the new work also needs to be discussed.

7) P2. Ln 25. The statement that 'From laboratory experiments it is established that dust particles tend to nucleate ice below -20C whereas biological particles can initiate immersion freezing at temperatures up to -5 C' is wrong. I can point to numerous studies showing dust can nucleate ice at much warmer temperatures; e.g. [Atkinson et al., 2013; Niemand et al., 2012; Ullrich et al., 2017]. Modelling suggests that dust

Interactive
comment

is important in many locations at much warmer temperatures than -20 C [Vergara-Temprado et al., 2017].

8) P2-3. It is not possible to distinguish between dust and bio INP on the basis of an inflection at -16oC. It is false to claim that such an inflection would give you information about biological INP. In making this statement the authors are assuming they know what the ice nucleating spectrum of dust is and also that they know that biological INP nucleate around -16C. Neither can be assumed or are correct. Biological material has a huge diversity in its nucleating ability. There are exceptional ice nucleating materials from specific fungal and bacterial species and much less active materials associated with marine biology. Also, the work of DeMott et al. [2016] and Wilson et al. [2015] suggest that the slope of INP vs T for marine INP materials is quite shallow, in contrast to what is stated here.

9) P4. Ln 5. It would be helpful to see the control fraction frozen curves as well as the fraction frozen curves for the samples. These control experiments look better than those reported by Conen et al., why is this? What has been done differently?

10) Figure 1. Also show other INP parameterisations that are used in models in addition to Fletcher, e.g. Meyers et al, Cooper et al.

11) P10, ln 15-23. In this discussion of the conclusion that the authors see no evidence for marine INP, they need to cite other papers with similar conclusions. For example, Fig 5 of Vergara-Temprado et al. [2017] clearly shows that desert dust is much more important than marine organic INP in the Eastern Atlantic region. Similarly to the final statement referring to Burrows, Wilson et al. [2015] also conclude that marine INP might be important in the southern ocean. They do not make this conclusion on the basis that marine organics are particularly good at nucleating ice, they conclude this because the southern ocean atmosphere has very little desert dust in it and marine organics therefore define INP population.

12) P11, ln 15, Why are INP above -10 biogenic? This statement needs to be expanded

upon or altered. As mentioned above, mineral dust can nucleate ice in this temperature regime. 13) P12. In this discussion of Ansmann et al., make it clear that this -20C number is for shallow clouds only, not deep convective clouds. In contrast the OSCIP from Rango and Hobbs is for cumulus clouds. Consequently I think the link between these ground level measurements and mid-level clouds is not as clear as the authors suggest.

14) Conclusions: Some of these paragraphs are very short and there is a single sentence paragraph which seems to be floating and not connected to other statements. Hence, it reads more like a list of bullet points than a well-crafted conclusions section. This could be improved.

References

Atkinson, J. D., B. J. Murray, M. T. Woodhouse, T. F. Whale, K. J. Baustian, K. S. Carslaw, S. Dobbie, D. O'Sullivan, and T. L. Malkin (2013), The importance of feldspar for ice nucleation by mineral dust in mixed-phase clouds, Nature, 498(7454), 355-358, doi:10.1038/nature12278.

Burrows, S. M., C. Hoose, U. Pöschl, and M. G. Lawrence (2013), Ice nuclei in marine air: biogenic particles or dust?, Atmos. Chem. Phys., 13(1), 245-267, doi:10.5194/acp-13-245-2013.

DeMott, P. J., et al. (2016), Sea spray aerosol as a unique source of ice nucleating particles, P. Natl. Acad. Sci. USA, 113(21), 5797-5803, doi:10.1073/pnas.1514034112.

McCluskey, C. S., et al. (2017), A Dynamic Link between Ice Nucleating Particles Released in Nascent Sea Spray Aerosol and Oceanic Biological Activity during Two Mesocosm Experiments, J. Atmos. Sci., 74(1), 151-166, doi:10.1175/jas-d-16-0087.1.

Niemand, M., et al. (2012), A particle-surface-area-based parameterization of immersion freezing on desert dust particles, J. Atmos. Sci., doi:10.1175/jas-d-11-0249.1.

Ullrich, R., C. Hoose, O. Möhler, M. Niemand, R. Wagner, K. Höhler, N. Hiranuma, H.

Saathoff, and T. Leisner (2017), A New Ice Nucleation Active Site Parameterization for Desert Dust and Soot, J. Atmos. Sci., 74(3), 699-717, doi:10.1175/jas-d-16-0074.1.

Vali, G., P. J. DeMott, O. Möhler, and T. F. Whale (2015), Technical Note: A proposal for ice nucleation terminology, Atmos. Chem. Phys., 15(18), 10263-10270, doi:10.5194/acp-15-10263-2015.

Vergara-Temprado, J., et al. (2017), Contribution of feldspar and marine organic aerosols to global ice nucleating particle concentrations, Atmos. Chem. Phys., 17(5), 3637-3658, doi:10.5194/acp-17-3637-2017.

Wilson, T. W., et al. (2015), A marine biogenic source of atmospheric ice-nucleating particles, Nature, 525(7568), 234-238, doi:10.1038/nature14986.

Yun, Y., and J. E. Penner (2013), An evaluation of the potential radiative forcing and climatic impact of marine organic aerosols as heterogeneous ice nuclei, Geophys. Res. Lett., 40(15), 4121-4126, doi:10.1002/grl.50794.

---

## Author Comment (AC1) · 18 Dec 2017

**Response to the Comments of Keith Bigg (Reviewer 1)**

We would like to thank Dr Bigg for his helpful comments and suggestions. We address the individual comments point by point below.

**General comment** *This manuscript provides a potentially valuable long series of IN data for an area from which there have been no previous IN measurements that I am aware of. The topic is suitable for ACP and the methods used appear to be satisfactory, although there is a suspicion that concentrations may be underestimated. The language and presentation are clear and the few changes that seem desirable will be listed at the end of this review. This is a good paper and I hope that some of the following comments might be helpful. Where the authors disagree with those comments, I would be happy to enter into a debate.*
*Summarising, I recommend publication after the authors have made any changes that appear to be appropriate.*

**Specific comments**

1. ***Abstract:*** *The paragraph that shows no relation between bulk chemical composition and IN concentrations is not very useful in the main text and certainly not in the Abstract. At a site where sea salt particles will be dominant by mass and IN active at temperatures relevant to cloud formation represent about one particle in a million, no other result would be credible.*

   We agree with Dr. Bigg that no direct correlation of the IN composition to the bulk aerosol composition is to be expected due to the rarity of IN in the total aerosol number concentration. The idea of the presented analysis is to use changes in bulk chemical composition of the aerosol as a tracer for different sources. This way, even if only one in a million particles acts as IN we expected to see a proportional increase in concentration with the increase of one of the "tracer materials" for different sources. The finding that this is not the case could indicate that none of these sources (marine, continental, combustion) is the strong supplier of IN active in this temperature range or that IN are distributed independently of their sources main compounds, or that additional conditions affecting the IN content in these sources have to be met. The result confirms that a simple attribution of an ice active particle fraction to a certain source is not feasible.
   The section was removed from the abstract. We decided to keep the part in the main text as we think the implications of comparing to bulk chemistry, support the findings in the section about the air mass origin.

2. ***Introduction:*** *p1., line 21. Is there any firm evidence for concentrations as low as $0.01 m^{-3}$ except in polar regions? In addition to the listed factors influencing the formation of precipitation, updraft velocities and liquid water content could be added. Cloud extent suggests horizontal extent – cloud depth would be better.*

   We could not find published measurements of such low ice crystal concentrations. The reason for the lack of measurements could be the lower detection limit of ~$0.1\ell^{-1}$ typical for instruments deployed for the task (e.g. 2D-S, Farrington et al., 2016). The description of the visual appearance of ice clouds containing different concentrations of ice crystals is removed from the manuscript because it was not essential.
   Updraft velocities and liquid water content are added to the list of factors influencing properties of glaciating clouds and "cloud extent" has been replaced by "cloud depth".

3. *p2 line 1: What concentration of ice crystals is necessary for ground-based detection? The presence of concentrations of IN in clouds at -10C will usually be similar to that of the concentrations in the air feeding the cloud of all biological IN capable of nucleation at that temperature. The list of potential IN from those sources continues to grow and concentrations must often exceed $100 m^{-3}$.*

   The lower limit of ice crystal concentrations detectable by ground based Lidar measurements are on the order of ~$0.5\ell^{-1}$ (A. Ansmann, personal communication). In-situ detectors like the 2D-S are sensitive to concentrations larger than ~$0.1\ell^{-1}$.

Sesartic et al. (2013) reviewed measurements of bacteria and fungal spore concentration in the atmosphere and modeled their impact on clouds. Concentrations can be more than $1000 m^{-3}$, but most bioaerosol seem to remain near their source of origin. Their presence is strongly coupled to vegetation and their ability to act as IN sometimes needs to be triggered by stress factors. It could be speculated that bacteria and fungi living in a subtropical habitat rarely encounter stress factors that they could counteract by expressing ice nucleating properties.

Because the importance of biological IN is circumstantial in the present study, no changes to the text were made in this context.

4. *P2. line 5. Insert reference here: Hallett and Mossop, Nature 249, 26-28, 1974 who were first to investigate the effect experimentally and define the conditions necessary for it to operate. Line 15 Hallett and Mossop defined that condition in 1974!*

We now refer to Hallett and Mossop (1974) and added to the manuscript: "Hallett and Mossop (1974) suggested that marine cumuli contain large ice crystal concentrations for dynamical reasons. They usually have higher cloud top temperatures, therefore the contact of ice and supercooled droplet occurs at temperatures favorable for splintering."

5. *P2. Line 19-22. In Bigg's 1973 paper, 3 years of continuous measurements in the Southern Ocean, south Indian Ocean and south Pacific were summarised. The measurements were later used by Schnell and Vali (J. Atmos.Sci., 33(8), 1554-1564, 1976) to show that the measurements revealed a strong dependence on biological productivity. Their interpretation of biological IN rather than dust as the main factors in the measurements is much preferable to Bigg's.*
*This work brings up an important point in relation to your manuscript. The ocean measurements were made with membrane filters that are known to undercount the concentration of IN in a salt-laden atmosphere, yet concentrations in the biologically productive zones were considerably higher than those reported in your manuscript. Chlorophyll measurements in the vicinity of the Cape Verde is reported by Ramos et al. to indicate strong biological activity and significant biological IN should become airborne by bubble bursting. Am I right in assuming that your method only gives a "yes-no" answer for the presence of an active IN at temperature T? If so it needs to be pointed out that actual concentrations may be higher.*

We prefer to include both, the marine biological and the long-range transported desert dust IN interpretation. The evidence for marine biological IN being more important than other substances for ice formation at low supercooling, remains circumstantial. It could also be that biological IN adhere to desert dust and are transported with it (Conen et al., 2011).

Biological particles exhibit a complex variety of dependencies of their activity as IN on environmental parameters. Research into the importance of organic marine aerosol, released from the sea surface microlayer by bubble bursting, has been conducted by Wilson et al. (2015). They reported a wide range of freezing temperatures, wherein samples from the Atlantic tend to freeze at lower temperatures than Arctic samples. The hypothesis would be, that marine bacteria in the subtropic ocean are less active IN than strains living at higher latitudes. See also comment 8 and 10.

The measurement method does give a "yes-no" answer for each of the 103 subsamples. For the interpretation of the measurement, we use a non-stochastic view of the nucleation process. Under this premiss, the assumption that only one IN causes freezing in a freezing droplet, is justified in Vali (1971). A possible effect of long sampling times to deactivate IN by blocking active sites, is already mentioned in the introduction. We are not aware of other reasons why the method would under predict concentrations. No changes were made to the text.

6. *p.3 lines 3-4. A spectacular example of the changes in IN concentrations that can occur was published in J. Meteorol. 15, 561-562, 1958. This was later interpreted to be due to related to enhanced biological populations resulting from heavy rain, with a proportion becoming airborne. (Atmos. Chem. Phys., 15, 2313-2326, 2015).*

The 20-30 day change in IN concentration following intense rainfall discussed in Bigg (1958); Bigg et al. (2015) is a spectacular example for the importance of biological IN under certain conditions, at places close to their origin. The fluctuations we refereed to in the manuscript are on shorter timescales. In Bigg (1961) you refereed to these events as "sudden onset storms". To include both, we changed the sentence to: "The concentration of potential IN measured

at a certain location and temperature can undergo large changes on timescales of days (Bigg, 1958) or hours (Bigg, 1961)." On Cape Verde rain events seem to be followed by prolonged low IN concentrations (at -8 °C). See Fig. 1 for a time series including precipitation events. Rain washes the IN out. This could indicate that biological IN active at this temperature are transported over long distances, eventually attached to larger particles like desert dust.

7. *p.3 line 11-12. Long sampling times aren't necessary. Membrane filters with pore sizes 0.45μm or larger can be sampled at >10l/min but sampling >300l leads to serious undercounts. For long-term measurements or simultaneous measurements at many sites, sampling at 300l/day avoided logistic difficulties but averaged out any short-term fluctuations.*

   The filter used in this study were collected with the intention to investigate the chemical composition of the aerosol at Cape Verde. We reused them to find out about the IN concentration. The sampling time of these filters was 1-3 days and before the time consuming IN-measurements, we estimated the concentration range for which the method would be sensitive to if we took subsamples of different sizes. The calculation showed that thanks to the long sampling time (large sampled air volume) we could use the smallest practical subsample size. This was important to minimize the background from the filter material of the subsample. Much shorter sampling times would not have allowed to use these filters for the measurement of IN concentrations. We agree that, when collecting filters with the purpose to measure IN concentrations, it would be better using the shortest possible sampling interval to reveal short-term fluctuations. We rephrased the sentence. "To measure the low concentration of IN, able to initiate immersion freezing above -20 °C, it is necessary to collect particles from a large enough air volume. The time required to collect this volume depends on the sample flow through the filter, which is influenced by its type (fiber or membrane), fiber or pore size, and the capacity of the pump. Long sampling times may be necessary."

8. *p3. Line 14. Reduction of the RH in the vicinity of a hygroscopic particle is a major factor. Allowing hygroscopic material from a $1m^3$ sample on a membrane filter to diffuse into an underlying wet filter, then drying the top filter and processing it, results in an IN count more than a factor of 2 higher than on a simultaneously sampled filter kept dry. (This work has not been published – use the information if you want to).*

   For the analysis of the filter samples we assume the IN are acting in the immersion freezing mode, residing in droplets. The method consists of immersing pieces of the filter in water drops and observe the freezing of those. The method is immune to water-vapour depletion effects and the water drops are large enough to prevent freezing point depression by the dissolved sea salt within the sensitivity of the temperature measurement. No changes were made to the text in this context.

9. *p.5, line2. At first I didn't understand this as all particles capable of forming IN at temperatures warmer than the test temperature will be activated. Does the answer lie in comment 5?*

   With decreasing temperatures the change in activity of single substances is steeper than the observed change in IN concentration. Consequently substances contribute only at high activity to the ambient IN spectrum (see response to comment 2 of Reviewer 2 and Appendix A).

10. *Figure 2. According to Ramos et al. chlorophyll is a maximum at the end of the year at Cape Verde. I don't see much evidence of a corresponding change in the -8C figure. As it is an important point in determining whether biological IN are effective at the site, running means of about 9 measurements at -8C shown on a diagram with a more extended scale might help. This procedure would be useful in reinforcing the surprising statement in lines 6 and 7 on p.8.*

    Concentration data at -8 °C for the years 2011, 2012 and 5 months of 2013 are shown in Fig. 1. The data show no clear signal corresponding to the maximum in chlorophyll, Ramos et al. found in Nov. and Feb. The increased IN-concentrations in Feb. 2011 and 2013 occurred several days after dust events. It has been reported that bacteria express ice nucleating properties as a response to environmental stresses. It can be speculated that during times of high biological productivity, bacteria have a good time as well and don't need to express ice nucleating properties.

    We didn't include Fig. 1 into the manuscript and no changes were made to the text.

11. *Bulk chemical composition. See comment 1.*

[Figure]

**Figure 1.** Time series of IN concentration at -8 °C is shown in red. Daily precipitation amount (right hand axis) is shown as blue bars.

We prefer to keep the section in the manuscript. See comment 1.

12. *Air mass origin. How many cases were involved in each of the 7 categories of figure 4? The sporadic rainy season from August-September would probably lead to much deeper atmospheric mixing at times and occasional scavenging of aerosol. How reliable are the 10-day trajectories to the site during that period?*

The individual data points composing the box plots are shown on the left hand site of each box. This data points are color coded according to the quarter of the year in which they are observed. The number of individual data points is given in Tab. 1.

**Table 1.** Number of cases per air mass category. Categories are: 1. African coastal, 2. Atlantic marine, 3. North American, marine and coastal, 4. North American and marine, 5. dust and Europe, 6. dust (e.g. Saharan and Sahel region), 7. coastal and Europe.

| Temperature | 1. | 2. | 3. | 4. | 5. | 6. | 7. |
|---|---|---|---|---|---|---|---|
| -8 °C | 74 | 11 | 19 | 40 | 47 | 55 | 30 |
| -12 °C | 111 | 12 | 28 | 54 | 76 | 93 | 44 |
| -16 °C | 109 | 13 | 28 | 51 | 73 | 92 | 41 |

How efficient wet deposition removes aerosol depends on rain intensity, raindrop size, aerosol size (Jung et al., 2013). The NAME model should be able to detect any deep atmospheric mixing (the UM met data should be able to track these events). Rainfall is a rare event on the Cape Verde island and often of very light intensity. The NAME model did the simulation for an inert tracer so as to detect the overall air mass transport, rather than act as a chemical transport model that tracks aerosol movement. An analysis separating the rainy season as suggested in comment 15 showed that IN concentrations do not generally change in this period.

13. *Frequency distribution, p.9 lines 7-9. Size distribution of particles produced by a common method frequently have a log-normal size distribution and this can be expected from a local source. An alternative to Ott's random dilution hypothesis might simply be preservation of the original distribution during transport.*

If the size distribution is preserved, then we would not expect an effect on the concentration of IN over time. The change in concentration can have two causes. A change in the ice nucleation activity of the aerosol (e.g. due to size) or a change

in abundance. The change in activity with particle size can account for up to two orders of magnitude in concentration (cf. Fig.1 in the response to Reviewer 2). A connection of changes in the size distribution with abundance seems plausible e.g. lower wind speed, transporting less and smaller particles. However, if this would lead to a log-normal frequency distribution is unclear. It could be an additional factor besides random dilution that causes the large variation in IN-concentration. At this point, Ott's hypothesis seems the simplest explanation for the observed log-normal distribution. No changes were made to the text in this context.

14. *p.10, line 6. What is the minimum concentration of ice crystals needed for the lidar observations to detect them? It might be better to replace "to start" with "to be detected".*

The sentence has been changed accordingly. The lower limit for Lidar to detect ice is in the range of $\sim 0.5\ell^{-1}$, see comment 3.

15. *Figure B1, p.14. It might be interesting to have a separate diagram for the "rainy" season of August and September and for the period of maximum productivity, October-December.*

The extended Fig. B1 is shown below. No trend emerges from the rainy or biological active season. Fig. 2 is now included in Appendix B instead of former Fig. B1.

[Figure]

**Figure 2.** Fig. B1 extended by boxplots of the rainy season (Aug. and Sep.) and the season of high biological productivity (Oct.-Dec.)

**Technical corrections**

*p.1 line 22 : change "cloud extend" to "cloud depth".*
*p.2 line 26. Change "consist to" to "consist of".*
*p.5 line 14 "exemplary shown". Change to "exemplified by". Change "year" to "years".*
*p.8. line 6. Change "effected" to "affected".*
*p.9 line 20. Change "to identify" to "identification of".*
*p.10. line 10. Change "to identify" to "in identifying". Line 19: change "must not" to "need not".*

Done.

**References**

Bigg, E. K.: A long period fluctuation in freezing nucleus concentrations, J. Meteorol., 15, 561 – 562, 1958.

Bigg, E. K.: Natural Atmospheric ice nucle, Sci. Prog., 49, 458, 1961.

Bigg, E. K., Soubeyrand, S., and Morris, C. E.: Persistent after-effects of heavy rain on concentrations of ice nuclei and rainfall suggest a biological cause, Atmos. Chem. Phys., 15, 2313 – 2326, doi:10.5194/acp-15-2313-2015, 2015.

Conen, F., Morris, C. E., Leifeld, J., Yakutin, M. V., and Alewell, C.: Biological residues define the ice nucleation properties of soil dust, Atmos. Chem. Phys., 11, 9643 – 9648, doi:10.5194/acp-11-9643-2011, 2011.

Farrington, R. J., Connolly, P. J., Lloyd, G., Bower, K. N., Flynn, M. J., Gallagher, M. W., Field, P. R., Dearden, C., and Choularton, T. W.: Comparing model and measured ice crystal concentrations in orographic clouds during the INUPIAQ campaign, Atmos. Chem. Phys., 16, 4945–4966, doi:10.5194/acp-16-4945-2016, 2016.

Hallett, J. and Mossop, S.: Production of secondary ice particles during the riming process, Nature, 249, 26 – 28, 1974.

Jung, C., Lee, S., Bae, S., and Kim, Y.: Minimum Collection Efficiency Particle Diameter during Precipitation as a Function of Rain Intensity, Aerosol Air Qual. Res., 13, doi:10.4209/aaqr.2012.09.0255, 2013.

Sesartic, A., Lohmann, U., and Storelvmo, T.: Modelling the impact of fungal spore ice nuclei on clouds and precipitation, Environ. Res. Lett., 8, 4029, 2013.

Vali, G.: Quantitative Evaluation of Experimental Results on the Heterogeneous Freezing Nucleation of Supercooled Liquids, J. Atmos. Sci., 28, 402 – 409, doi:10.1175/1520-0469(1971)028<0402:QEOERA>2.0.CO;2, 1971.

Wilson, T., Ladino, L., Alpert, P., Breckels, M., Brooks, I., Browse, J., Burrows, S., Carslaw, K., Huffman, J., Judd, C., Kilthau, W., Mason, R., McFiggans, G., Miller, L., Najera, J., Polishchuk, E., Rae, S., Schiller, C., Meng, S., Vergara-Temprado, J., Whale, T., Wong, J., Wurl, O., Yakobi-Hancock, J., Abbatt, J., Aller, J., Bertram, A., Knopf, D., and Murray, B.: A marine biogenic source of atmospheric ice-nucleating particles, Nature, 525, 234 – 238, doi:10.1038/nature14986, 2015.

---

## Author Comment (AC2) · 18 Dec 2017

**Response to the Comments of Reviewer 2**

We would like to thank Reviewer 2 for his comments and thoughtful suggestions. We reply to the individual comments below.

**General comment** *First of all, this is a very impressive set of measurements. The field of atmospheric ice nucleation is lacking long term data sets with which to compare models and test our understanding. Hence, a dataset comprised of 500 individual measurements over the course of 5 years is extremely valuable. Hence, I support its publication. However, I think there are some aspects of the paper which need to be improved prior to publication. I go through these in detail below:*

**Specific comments**

1. *P1. Ln 11. Why does an exponential change in INP concentration suggest that several groups of particles with different ice nucleating properties are contributing to INP populations? The exponential dependency on T could also just be explained by a distribution of sites across a surface, rather than particles of different composition or size.*

   The crucial point in our view is that the INP concentration is observed to exponentially increase by seven order of magnitude from -5 °C to -38 °C i.e. the steepness with which the concentration increases on a log-scale. This increase is shallower than the increase in ice nucleation activity of any pure substance, bringing us to the conclusion that several substances contribute to the ambient temperature spectra. To clarify, we rephrase in the manuscript: "Concentration of ice nuclei is found to increase exponentially by seven order of magnitude from -5 °C to -38 °C. Sample to sample variation in the steepness of the increase indicates that particles of different origin, with different ice nucleation properties (size, composition) are contributing to the ice nuclei concentration at different temperatures."

   Our interpretation of the temperature spectra is a central point of criticism of Reviewer 2, also in comment 7, 8 and 12. We attempted to illustrate our point of view in Appendix A in the manuscript. Below we try to clarify and support our conclusions by following the Reviewers suggestion to use active surface site distributions to explain the observed temperature spectrum. We show how the atmospheric INP spectrum compares to an INP spectrum that is constructed using concentrations of ice nucleation active surface sites (referred to as INAS or $n_s$) for desert dust and for Microcline. The temperature spectra (INP concentration as function of temperature) generated by a substance is the substance's ice nucleation activity (e.g. frozen fraction as function of temperature) multiplied by the number of particles of this substance. We assume one particle per droplet.

   Frozen fractions can be derived from $n_s$ parametrizations by $FF = 1 - exp(-n_s \cdot A)$ with $A$ being the particle surface area per droplet. The frozen fraction envelop in Fig. 1(a) is obtained by using $A$ of spherical particles with diameter between 100nm and 1$\mu$m.

   Fig. 1(a) shows the calculated frozen fraction for $n_s$ parametrizations, found for the K-feldspar, Microcline (Atkinson et al., 2013) and different desert dusts (Ullrich et al., 2017). Microcline is considered the most ice nucleation active dust identified so far (Atkinson et al., 2013), we come back to that in comment 7. Desert dust is not a pure substance but a mixture of different minerals and whatever sticks to the mineral dust particles (e.g. biological residues, Conen et al., 2011).

   Looking at Fig. 1(a) it can be seen that the ice nucleation activity of both, desert dust and Microcline is higher than that of a hypothetical substance that would correspond to Fletcher's curve. As pointed out by the Reviewer, using $n_s$ results in an exponential dependency on temperature. However the slope of Microcline is steeper than the Fletcher curve. Based on laboratory measurements on other, seemingly pure substances (e.g Fig.1a, Atkinson et al., 2013) we argue that all pure mineral dusts exhibit a comparable or steeper slope than Microcline, i.e. activity increases more rapidly with decreasing temperature than that of a hypothetical substance that would correspond to the Fletcher curve. As a consequence, if only one mineral dust would contribute to the concentration of INP in the entire temperature spectra, the spectrum must have the same steepness as the frozen fraction of that mineral. Only Microcline, as an example, would produce the temperature spectrum shown in Fig. 1(a) with the right hand y-axis (calculated by $FF_{Microcline} \cdot N_{IN@233K}$). While INP concentrations between 263-268K would correspond to observational data, at 253K where typical INP concentration is ~$1\ell^{-1}$,

Microcline would produce ~$1000\ell^{-1}$. This is not what is observed. The lower temperature dependence of the frozen fraction calculated using $n_s$ of desert dust must result from a superposition of several steeper curves of all substances composing desert dust in their specific fraction.

Fig. 1(b) shows the temperature spectrum from multiplying desert dust's ice nucleation activity with a globally averaged dust concentration given in Atkinson et al. (2013). This represents observations well, in steepness and range. As mentioned above, we argue that the match in steepness stems from the contribution of different substances contained in desert dust.

[Figure]

**Figure 1.** (a) Frozen fraction (left side axis) of of Desert dust, Microcline and ambient INP as function of temperature. Frozen fractions are calculated based $n_s$ from Atkinson et al. (2013); Ullrich et al. (2017) for particle diameter between 100nm and $1\mu m$, respectively by normalizing Fletcher's parametrization by the maximum predicted INP concentration (at 233K). Multiplying frozen fractions by the normalization factor leads to the corresponding ice nuclei concentrations (right hand axis).
(b) Temperature spectra including Fletcher's parametrization and $n_s$ predicted concentration, scaled by globally averaged dust concentration for desert dust and 10% of globally averaged dust concentration for Microcline.

2. *Consider using the term 'ice nucleating particles; INPs' rather than ice nuclei (IN). Vali et al. [2015] in their recent definitions paper came up with some compelling reasons why this is a better and less confusing term.*

   As we are investigating filter samples and ice nuclei may be of various nature (particles, macro-molecules, ...) we rather prefer to stick to the classic abbreviation IN. We now state in the introduction that IN stands for "ice nucleating substance".

3. *P1 ln 24. Provide a citation for 100 L-1 INP leading to diamond dust. My understanding was that diamond dust was a relatively low concentration of ice crystals of relatively large size, i.e. in contradiction to the statement made here. In addition, I understand diamond dust tends to form in clear air, without the presence of a liquid cloud.*

   The sentence is inspired by the following report by Vincent Schaefer: "One late winter day in 1944, when I was climbing Mt. Washington by way of Tuckermans Ravine with Dr. Irving Langmuir, we approached the base of an orographic cap cloud which covered the cone of the mountain. He pointed out that the concentration of ice nuclei in the cloud was probably less than one per 1000 cubic meters, i.e. 1 x $10^{-6}$/liter! He made his estimate using the horizontal and vertical distances of about 10 meters existing between the snow pellets which were falling from the cloud. We were also aware that at other times, what we call diamond dust, could be observed under similar temperature conditions with

concentrations as high as 100 per liter." (Schaefer, 1967).

The Reviewer is correct that the connection to ice formation in liquid clouds is not obvious without context. As no firm evidence could be found elsewhere, the sentence is removed from the manuscript.

4. *P2, ln 1. When discussing data like that of Ansmann et al. and making statements such as 'above -10 C ice containing clouds are rare', make sure it is stated what sort of clouds are being referred to. For example, in convective clouds ice formation above -10oC is common. Ansmann et al. deal with shallow clouds.*

Ice formation via immersion freezing should occur at the same temperatures, independent on the dynamical conditions under which clouds form. The difference must be due to secondary ice formation, which is more effective in cumuli than in stratiform clouds.

The cloud type in the studies cited is added to the manuscript.

5. *P2. Ln 3. This paragraph is very confused. Parts of it seem to be referring to ice formation in shallow cloud types (e.g. stratus), whereas it then morphs into a discussion about secondary production which is more relevant for deep clouds.*

We have restructured and extended this part of the introduction to make it more organized. The cloud regime in which the measurements we refer to have taken place is indicated.

It now reads: "Ambient measurements of IN-concentrations from various studies were compiled by Fletcher (1962) to derive a spectrum of the average IN-concentration as a function of temperature (curve shown in Fig. 1). Fletcher's curve has been found to match ice crystal concentrations measured in stratiform clouds and cold-based convective clouds but underpredict the concentration in deep convective clouds (e.g. Mossop, 1985; Cooper, 1986). Matching concentrations of ice crystals and IN indicate a direct influence of immersion freezing IN on cloud properties.

At what concentration ice crystals exert a substantial influence on the properties of the cloud in which they form, has been addressed by Rangno and Hobbs (1988). From aircraft and mountaintop observations, Rangno and Hobbs (1988) identified the significant ice crystal concentrations able to produce precipitation to be on the order of $1 \, L^{-1}$ or more, cf. Fig. 1. In their data-set, such concentrations have been measured in cumulus clouds with top temperatures between -5 °C and -10 °C. At this temperature, this is a much larger concentrations than expected from the Fletcher curve and indicates secondary ice formation. Secondary ice formation mechanisms (most efficient at -5 °C) can increase the ice crystal concentration within a cloud by as much as four orders of magnitude above the number of IN present (e.g. Hallett and Mossop, 1974; Hobbs and Rangno, 1985; Mossop et al., 1970). Already few active IN ($0.01 \, L^{-1}$) can be enough to start the multiplication (Sullivan et al., 2017). In-situ evidence for secondary ice formation is reported in Hoffer and Braham (1962). They collected graupel from the top of cumulus clouds to melt and refreeze them under laboratory conditions. They found that every sample froze at substantially lower temperatures than the lowest temperature in the cloud from which they were collected, indicating that they did not contain IN active at the temperature the ice pellets froze in the cloud.

Typical temperatures at which ice crystals are observed to form in numbers high enough to affect the properties of clouds have been measured by remote sensing. Satellite observations, averaging clouds globally (e.g. Carro-Calvo et al., 2016) and ground-based Lidar measurements, looking at shallow clouds (e.g. Ansmann et al., 2009), report that above -10 °C ice containing clouds are rarely detected and often cloud top temperatures below -20 °C are necessary for clouds to glaciate. Satellite data (Carro-Calvo et al., 2016; Rosenfeld et al., 2011) and aircraft observations (Rangno and Hobbs, 1991, 1994) agree on a land-sea contrast with the tendency of cloud glaciation at higher cloud top temperatures over sea. Carro-Calvo et al. (2016) offer the explanation that the presence of larger sized droplets in maritime clouds, which are required for effective secondary ice formation (Heymsfield and Willis, 2014) could play a role. Hallett and Mossop (1974) suggested that marine cumuli contain large ice crystal concentrations for dynamical reasons. They usually have higher cloud top temperatures, therefore the contact of ice and supercooled droplet occurs at temperatures favorable for splintering."

6. *P2. Ln 17-25. This discussion of marine INP is lacking reference to some more recent literature on the subject, e.g.: [Burrows et al., 2013; McCluskey et al., 2017; Vergara-Temprado et al., 2017; Wilson et al., 2015; Yun and Penner, 2013]. I appreciate the effort made to go back to much older studies, but the new work also needs to be discussed.*

We thank the Reviewer for pointing us to this recent studies. We included some of them at this point of the manuscript. "More recently, laboratory investigations (DeMott et al., 2016; McCluskey et al., 2017), modeling (Burrows et al., 2013; Vergara-Temprado et al., 2017) and ambient observations (Wilson et al., 2015) showed that under certain conditions, IN from marine sources can be abundant enough to significantly contribute to the total IN population, particularly in high latitudes."

7. *P2. Ln 25. The statement that 'From laboratory experiments it is established that dust particles tend to nucleate ice below -20C whereas biological particles can initiate immersion freezing at temperatures up to -5 C' is wrong. I can point to numerous studies showing dust can nucleate ice at much warmer temperatures; e.g. [Atkinson et al., 2013; Niemand et al., 2012; Ullrich et al., 2017]. Modeling suggests that dust is important in many locations at much warmer temperatures than -20 C [Vergara-Temprado et al., 2017].*

We changed the sentence to:'From laboratory experiments it is established that dust particles tend to nucleate ice efficiently below -20 °C whereas some biological particles can efficiently initiate immersion freezing at temperatures up to -5 °C (Murray et al., 2012).'

Modeling INP concentrations with only Feldspar suffers on over-predicting concentrations at low temperatures and under-predicting concentrations at warm temperatures, in agreement with what is demonstrated in Fig. 1, comment 1. The temperature of agreement between model output and observation shifts with the feldspar number concentration.

8. *P2-3. It is not possible to distinguish between dust and bio INP on the basis of an inflection at -16oC. It is false to claim that such an inflection would give you information about biological INP. In making this statement the authors are assuming they know what the ice nucleating spectrum of dust is and also that they know that biological INP nucleate around -16C. Neither can be assumed or are correct. Biological material has a huge diversity in its nucleating ability. There are exceptional ice nucleating materials from specific fungal and bacterial species and much less active materials associated with marine biology. Also, the work of DeMott et al. [2016] and Wilson et al. [2015] suggest that the slope of INP vs T for marine INP materials is quite shallow, in contrast to what is stated here.*

There might be a misunderstanding. As shown in Fig. 1 comment 1, mineral dust contribute at temperatures lower than -16 °C, therefore changes in abundance of dust shifts the temperature spectrum up and down, below this temperature. Any inflection at warmer temperatures can be attributed to other INP species (which don't need, but could be biological). It is more difficult to differentiate between them on the basis of the temperature spectrum at lower temperatures.

Shallow slope of any material must be due to its inhomogeneity. The largest inhomogeneity in marine INP is probably the diversity in composition i.e. various particle species in different concentrations and with differing activities. Laboratory results pointing in this direction can be found in McCluskey et al. (2017). They report that experiments when a phytoplankton bloom was provoked, caused a response at various temperatures during different stages of the bloom, suggesting a diverse marine INP population.

The section was slightly changed to make it clearer. It now reads:"Two main sources for IN in maritime air have been proposed. Long-range transported continental aerosol (mainly dust) suggested by Bigg (1973) or marine organic ice nuclei of biogenic origin aerosolized with the sea spray (Schnell and Vali, 1975; Rosinski et al., 1987). Aerosol from both sources have been investigated (e.g. DeMott et al., 2003, 2016, and references therein). They found that marine-sourced IN are less efficient than IN from continental sources and can contribute IN in a broad temperature range. From laboratory experiments it is established that dust particles tend to nucleate ice efficiently below -20 °C whereas some biological substances can efficiently initiate immersion freezing at temperatures up to -5 °C (Murray et al., 2012). Joly et al. (2014) demonstrated that the particles initiating immersion freezing in cloud water samples collected at Puy de Dôme consist to an increasing fraction of biological IN (identified by sensibility to heat treatment) towards higher freezing temperatures. They estimate 77% biological IN at -12 °C increasing to 100% at -8 °C.

The different temperature range in which dust or biological particles act efficiently as IN, can generate a specific signature in the concentration-temperature spectra if there is a change in abundance (cf. Sec. 4 and Appendix A). Some biological particles and in particular bacterial IN of one source tend to exhibit homogeneous ice nucleating properties. They initiate ice formation in a narrow temperature range seen as step like increase in concentration in a temperature spectra (e.g. Murray et al., 2012). In contrast, dust particles activate in a broader temperature range (e.g. due to inhomogeneities in

composition, surface structure and the influence of particle size on nucleation efficiency) seen as exponential increase in concentration towards lower temperatures (Bigg, 1961). If a strong marine source of biological origin exists it can be expected to be detectable as an inflection in IN-concentration at temperatures above -16 °C where the ice nucleating fraction of dust particles is small."

9. *P4. Ln 5. It would be helpful to see the control fraction frozen curves as well as the fraction frozen curves for the samples. These control experiments look better than those reported by Conen et al., why is this? What has been done differently?*

We don't know what makes the difference. Fig. 2(a) shows the frozen fraction curves from the reported samples in comparison to three measurements of each the used water, fresh filter and field blanks of filters treated, stored and handled the same way as samples but removed from the Digitel before sampling. The contribution from sampled aerosol to the frozen fraction can be isolated from the background contribution of water and filter material. According to the 'At least one Rule' in probability calculus, freezing is caused by the aerosol, the background or both. Treating the frozen fraction as probability for a droplet to freeze we can calculate the probability for a droplet not to freeze:

$$(1 - FF_{BG}) \cdot (1 - FF_{Aerosol}) = 1 - FF_{Sample} \tag{1}$$

Solving for $FF_{Aerosol}$ results in

$$FF_{Aerosol} = \frac{FF_{BG} - FF_{Sample}}{FF_{BG} - 1} \tag{2}$$

where $FF_{BG}, FF_{Aerosol}, FF_{Sample}$ denoting the frozen fraction only from the background, the aerosol and the measured frozen fraction of the sample. With $FF_{BG}$ and $FF_{Sample}$ known, Eq. 2 can be used to calculate the contribution which can only be attributed to the sampled aerosol. The result is shown in Fig. 2(b).

[Figure]

**Figure 2.** (a) Frozen fraction curves of filter samples ($FF_{Sample}$) in comparison to background measurements. (b) Minimal frozen fraction from sampled aerosol ($FF_{Aerosol}$). The $FF_{BG}$ derived as a fit to blank measurements is shown in green. $FF_{Aerosol}$ curves are cut if their slope becomes negative and endpoints are marked in orange.

For some measurements, isolating $FF_{Aerosol}$ in the way described above can lead to decreasing frozen fraction with decreasing temperatures. As this is unphysical, we cut such curves at their inflection point.
The above analysis supports the temperature information given in the manuscript. We did not make changes to the text.

10. *Figure 1. Also show other INP parameterisations that are used in models in addition to Fletcher, e.g. Meyers et al, Cooper et al.*

The two parametrizations from Cooper (1986); Meyers et al. (1992) have been added. In order not to overload the figure and distract from the actual measurements, no other additional parametrizations are shown. The temperature range of the

parametrization lines has been capped according to the validity range given in the original work. The parametrization lines are calculated according to Eqs. 3, 4, 5 and shown in the range of supercooling ($\Delta T$) indicated:

$$\text{Fletcher 1962:} \quad N_{IN}[m^{-3}] = 10^{-2} \cdot exp(0.6 \cdot \Delta T), \qquad\qquad 10° < \Delta T < 30° \quad (3)$$

$$\text{Cooper 1986:} \quad N_{IN}[m^{-3}] = 10^3 \cdot 10^{(-2.35-0.135\cdot-\Delta T)} = 10^{(0.65+0.135\cdot\Delta T)}, \qquad 5° < \Delta T < 25° \quad (4)$$

$$\text{Meyers et al. 1992:} \quad N_{IN}[m^{-3}] = 10^3 \cdot exp(-0.639 + 0.1296 \cdot (100 \cdot (S_i - 1)))$$

$$= 10^3 \cdot exp(-0.639 + 0.1296 \cdot (100 \cdot (S_w \cdot \frac{p_{w,sat}(T)}{p_{i,sat}(T)} - 1))), \quad 7° < \Delta T < 20° \quad (5)$$

11. *P10, ln 15-23. In this discussion of the conclusion that the authors see no evidence for marine INP, they need to cite other papers with similar conclusions. For example, Fig 5 of Vergara-Temprado et al. [2017] clearly shows that desert dust is much more important than marine organic INP in the Eastern Atlantic region. Similarly to the final statement referring to Burrows, Wilson et al. [2015] also conclude that marine INP might be important in the southern ocean. They do not make this conclusion on the basis that marine organics are particularly good at nucleating ice, they conclude this because the southern ocean atmosphere has very little desert dust in it and marine organics therefore define INP population.*

   We added Vergara-Temprado et al. (2017).

12. *P11, ln 15, Why are INP above -10 biogenic? This statement needs to be expanded upon or altered. As mentioned above, mineral dust can nucleate ice in this temperature regime.*

   In the course of improving the conclusion section the statement was deleted. However, we think there is evidence that it is true. See comment 1. We conclude this based on the observed INP-concentration, which is the product of activity and abundance of the substance. The activity of mineral dust at -10 °C is low and a high number of particles would be necessary to produce the INP-concentration observed in the atmosphere. At lower temperatures, the high abundance of dust particles would result in much more INP than observed (see the calculation example in comment 1 and Fig. 1b). This leads to two possible explanations. First, the explanation offered in our paper, that what makes the temperature spectra are the concentrations of different substances contributing in a temperature range where their activity is high . As the most active mineral dust known today is not active enough to explain the observed INP concentration at $T > -20$ °C, another substance must be nucleating ice. Because Microcline is the most active mineral dust, no other mineral dust can be this more active substance. Natural substances with a higher activity are found in the class of biological aerosol. Alternative, there is a more active, very rare (low concentration) mineral dust that has not jet been identified or an entirely different source.

13. *P12. In this discussion of Ansmann et al., make it clear that this -20C number is for shallow clouds only, not deep convective clouds. In contrast the OSCIP from Rango and Hobbs is for cumulus clouds. Consequently I think the link between these ground level measurements and mid-level clouds is not as clear as the authors suggest.*

   Ansmann et al. (2009) measured alto cumulus clouds. This is now mentioned in the manuscript. The evidence is circumstantial, but the point is that an ice crystal concentration of $1\ell^{-1}$ is a benchmark concentration to clearly change mixed-phase cloud properties. In clouds where primary ice nucleation dominates ice crystal formation, this concentration is observed at -20 °C in agreement to our measurement of the INP-spectra at ground-level. We think this is an interesting agreement and kept it in the text. It could also be evidence that concentrations measured at ground are similar at cloud level.

14. *Conclusions: Some of these paragraphs are very short and there is a single sentence paragraph which seems to be floating and not connected to other statements. Hence, it reads more like a list of bullet points than a well-crafted conclusions section. This could be improved.*

We re wrote and condensed parts of the conclusion. It now reads:

[revised manuscript text omitted]

---

## Author Response (AR2)

**Response to the Comments of Allan Bertram (Handling Editor)**

We thank Dr Bertram for his comments and address them point by point below.

**Comments**

1. *P3 line 3. Should "consist" be changed to "contributed" and should "sensibility" be change to "susceptibility"?*
   We changed "sensibility" to "susceptibility", but kept "consist".

2. *P3. line 11. Consider changing "it can be expected to be detectable" to "it may be detectable".*
   Done.

3. *P5. line 13-15. I think this statement assumes that sigma in the alpha-pdf model is relatively small. A sentence mentioning this assumption would be useful at this point.*
   The statement holds for rather large sigma values ($\sigma = 6°$ in the example shown in Appendix A, which is on the higher end for mineral dusts). Larger sigma values indicate a mixture of particles (e.g. ATD, $\sigma \approx 9°$). To clarify we added to the statement: "Assuming that IN of certain size and composition develop their ice nucleation activity in a narrow temperature range, a low activated fraction of an abundant IN at one temperature would generate a steep increase in concentration with decreasing temperature (cf. Appendix A for an extended discussion on the interpretation of IN temperature spectra)."

4. *P11. line 19. Check spelling.*
   We changed "time-serie" to "time-series".

5. *P12. line 11. I think "given" should be changed to "assuming" unless there is evidence suggesting concentrations of INPs at the surface are the same as concentrations of INPs at higher altitudes.*
   Done.

The marked-up manuscript version with changes is attached below.

[revised manuscript text omitted]